# GC-MS Silylation Derivative Method to Characterise Black BIC^®^ Ballpoint 2-Phenoxyethanol Ratio Evaporation Profile—A Contribution to Ink Ageing Estimation

**DOI:** 10.3390/molecules28124781

**Published:** 2023-06-15

**Authors:** Teresa Argente Leal, Carla Ferreira, Ana Ribeiro, Samir Marcos Ahmad, Alexandre Quintas, Alexandra Bernardo

**Affiliations:** 1Forensic and Psychological Sciences Laboratory Egas Moniz, Campus Universitário—Quinta da Granja, Monte da Caparica, 2829-511 Caparica, Portugal; 2Molecular Pathology and Forensic Biochemistry Laboratory, Centro de Investigação Interdisciplinar Egas Moniz (CiiEM), Instituto Universitário Egas Moniz (IUEM), Campus Universitário—Quinta da Granja, Monte da Caparica, 2829-511 Caparica, Portugal

**Keywords:** GC/MS, viscous pens, gel pens, dating, volatile solvents, 2-phenoxyethanol, ageing curve

## Abstract

One of the major challenges in forensic document analysis is estimating the age of ink deposition on a manually written document. The present work aims to develop and optimise a methodology, based on the evaporation of 2-phenoxyethanol (PE) over time, that can contribute to ink age estimation. A black BIC^®^ Crystal Ballpoint Pen was purchased in a commercial area, and ink deposition began in September 2016 over 1095 days. For each ink sample, 20 microdiscs were subjected to n-hexane extraction in the presence of an internal standard (ethyl benzoate) followed by derivatisation with a silylation reagent. A gas chromatography-mass spectrometry (GC/MS) method was optimised for PE-trimethylsilyl (PE-TMS) to characterise the ageing curve. The developed method presented good linearity between 0.5 and 50.0 μg mL^−1^, as well as limits of detection and quantification of 0.026 and 0.104 μg mL^−1^, respectively. It was possible to characterise PE-TMS concentration over time, which reveals a two-phase decay behaviour. First, there was a substantial decline between the 1st and the 33rd day of deposition, followed a by a stabilisation of the signal, which allowed to detect the presence of PE-TMS up to 3 years. Two unknown compounds were also present and allowed to identify three dating time frames for the same ink stroke: (i) between time 0 and 33 days, (ii) between time 34 and 109 days, and (iii) more than 109 days. The developed methodology allowed to characterise the behaviour of PE over time and to establish a relative dating of three-time frames.

## 1. Introduction

One of the criminal activities with a significant impact on transnational crime and terrorism is the falsification or counterfeiting of documents [1]. Questioned document analysis is the primary forensic approach to verify erasures, added material, and writing obliteration in documents, providing crucial information for crime investigation.

Age determination of a document is among the major forensic challenges, but it is also key to providing the chronology of events for document production or signatures deposition on a paper. A document may consist of several entries made at different times, and questions may arise about whether a particular entry was made at a purported time. The expert analysis of these documents that can provide information on the inks’ deposition date is complex and usually involves advanced chemical and physical analysis [2,3,4].

The ink degradation mechanism is a complex combination of various processes. The natural ageing of paper, as well as the ink composition and its capacity to produce chemical reactions, can contribute to this degradation process. The environmental and storage conditions can significantly influence these chemical reactions. These processes can be simple (e.g., solvent evaporation) or complex (e.g., hardening of the ink resins on the paper surface). For paper, the ageing process starts after its manufacture, whereas for ink it begins after it is affixed on paper, including the evaporation of volatile compounds. Indeed, after their manufacture, inks are stored in containers (pen cartridges and ink bottles), and their ageing in these containers is negligible compared with the ageing which starts as soon as the ink is affixed on paper (open system). On the other hand, the ageing of dyes starts when the ink is placed in the pen cartridge because of their exposure to light which result in their degradation [5].

Among the methods to date documents through ink analysis already developed, two main approaches can be distinguished: a static and a dynamic. The static approach performs an analytical chemical profile of the ink’s stable compounds, and the dynamic approach determines the chemical profile of the ink’s time transient compounds, after deposition on paper, hence beginning its contact with air, light, and relative humidity. For document ageing determination, the dynamic approach is the most common choice and includes the analysis of colorant decomposition, evaporation and diffusion of solvents, and resin polymerisation. Colourants decompose gradually over time producing a wider time window for age determination. Solvents stabilise within 2 or 3 years, and the polymerisation of the resin becomes stable between 8 months and 2 years. In addition to the two approaches described (static and dynamic), several studies report supplementary approaches that complement the results obtained. These approaches include accelerated ageing techniques, chemometrics, construction of document chronology, luminescent component analysis, radiocarbon dating, and nanotechnology-based methods [3,6,7]. In the literature, the case reports show that the ageing document determination is mainly performed by the determination of the changes in crystal violet colorant and the solvent 2-phenoxyethanol (PE) [8,9,10,11].

The ink dating by dye behaviour characterisation has recently been explored by supplementary non-destructive techniques such as: (i) Raman spectroscopy applied to natural and artificial aging study of writing inks printed on paper [12] and (ii) Digital Colour Analysis [13], where the authors achieved a superior precision throughout an ink classification according with parameters of their colour degradation curves when compared with conventional Raman scattering method.

Concerning solvent ink ageing approach, several destructive analytical techniques such as GC–MS, GC-FID, FTIR, SPME, SPME-GC–MS, HS-GC-FID, and DART-MS provide information tracing the ink changes throughout evaporation profiles of volatile components [14,15,16].

According to a recent Interpol report, the most studied method is the analysis by GC/MS of PE [11]. This compound is a colourless, oily liquid with a faint rose-like odour that is commonly used as a preservative in cosmetic products, including lotions, shampoos, and makeup, and can be found in 85% of black inks and 83% of blue inks [17,18]. These two ink colours are the most common in pens used in official documents. Several groups have been developing approaches focused on PE aiming to find the most reliable analytical tool to date documents. PE is a standard pen solvent that is relatively stable and does not degrade rapidly over time, making it a useful marker for analysing relatively recent ink deposition in documents. The fundamental idea has been to determine ageing curves showing the decrease of this solvent in the ink entry with time. In the last two decades, several analytical methodologies were explored to analyse PE behaviour, namely: (i) thermal desorption-gas chromatography/mass spectrometry (TE-GC/MS) [14,19], (ii) solid–liquid extraction (SLE) followed by GC/MS [5,20,21], and (iii) solid phase microextraction followed by GC/MS [22].

Concerning the advances in the thermal desorption-gas chromatography/mass spectrometry (TE-GC/MS) technique, the results of the cross-laboratory validation revealed that ageing kinetics is comparable between instruments over the studied duration (x years) and should be considered instead of absolute values of ageing parameters whose repeatability of measurement is reliable only over a short period of time, even when analysing samples of the same paint stored under the same conditions. On the other hand, this technique also revealed that the ageing curves change when the storage conditions are modified and that it is necessary to verify if the results of the method chosen for data treatment depend on the number of data introduced for treatment. Overall, TE-GC/MS has shown that the variation of results obtained using this methodology is not only due to analytical limitations. Limitations, such as not knowing the storage conditions or the inhomogeneity of the paint inputs, are some of the characteristics of all forensic samples and condition the interpretation of the results from a legal perspective [9].

The phenoxy ethanol evaporation behaviour over time analysis after solid–liquid extraction, followed by GC/MS analysis, showed promising preliminary results in determining the chronological age of ballpoint ink when compared with other methods such as IR, transmittance, luminescence, and UV excitation [4].

Finally, when solid phase microextraction analysis followed by GC/MS is used to analyse PE behaviour, the results were revealed to be consistent, and the technique required minimum sample manipulation and can be applied to detect whether inks are less than 6 months old. However, this technique showed some drawbacks such as dependence on the type of paper, which cannot be ignored when different papers need to be compared. Moreover, the intensity of 2-phenoxyethanol detected becomes very faint after c. 1 year [22].

As described in the previously mentioned studies, the number of variables influencing the ink–paper system is considerable, and no solution has yet been found to achieve a suitable method for routine use in a forensic laboratory since most are limited to the analysis of PE and do not take into consideration ratio profiles during ink ageing.

Actually, dating deposited inks implies knowledge and control of variables of different natures where care in the handling of the instrument and of the samples and the validation of techniques between laboratories followed by blind tests on real samples are fundamental and where the probabilistic data processing may provide answers in a legal context [7]. Moreover, ink dating remains a challenge [11] where the contribution of different analytical approaches is crucial to achieving a validated method. The present work contributes to the literature with a proof of concept to date documents based on PE time profile behaviour, where the SLE extract is derivatised prior to GC/MS analysis. This step can improve method sensitivity to extend the period of PE detectability, distinguish small differences in PE decay curves, and enable the analysis of other compounds that present low volatility or are thermolabile.

## 2. Results and Discussion

### 2.1. GC/MS Results

First, we established the GC/MS conditions for separation and detection of ink constituents. This included an initial derivatisation step with a silylation reagent, since pen inks usually have a heterogenous mixture of compounds with different physico-chemical properties, including dyes, pigments, stabilising polymers, liquid solvents, and other additives [8]. In this case, this procedure is crucial for (i) improvement of the resolution and reducing the tailing of polar compounds, (ii) analysis of relatively non-volatile compounds, (iii) improvement of analytical efficiency and increasing detectability, and (iv) stabilising compounds for GC/MS analysis [8]. After applying the proposed approach, it was possible to identify in the black point pen ink three target peaks with retention times of 22.7 min (peak 1), 23.1 min (PE-TMS, peak 2), and 23.8 min (peak 3), Figure 1. We also confirmed that they were not present in the blank samples (paper). Moreover, the developed method was linear for PE-TMS between 0.5 and 50.0 μg mL^−1^, resulting in a R^2^ of 0.9995. The LOD and LOQ were 0.026 and 0.104 μg mL^−1^, respectively.

Although we only had the standard solution for PE, the obtained mass spectra peaks #1 and #3 suggest that they correspond, respectively, to 2-phenoxypropanol-TMS and 1-phenoxypropan-2-ol-TMS. Their mass spectrum and fragmentation patterns can be found in Figure 1b 2-phenoxypropanol-TMS and Figure 1d 1-phenoxypropan-2-ol-TMS, respectively.

Although, we could not find evidence in the literature, possibly due to the confidential nature of the pen inks formulation, we believe that we successfully identified these compounds since they present a similar structure to PE. Nevertheless, the identity of the compounds is not crucial to achieving the goals of the present work.

### 2.2. Ink Ageing Studies

To understand the decay of PE over time of the black BIC^®^ ballpoint pen ink, an ageing curve was generated between the ink depositions on the paper from day 0 to the day 1095 and the ratio between the PE and internal standard areas. The ageing curve showing two different behaviours. Thus, a two-phase exponential decay function was successfully fitted to the PE evaporation ageing dynamics (Figure 2), suggesting a fast and a slow slope in the ageing curve.

The two regions can be explained through the physicochemical processes that occur in the components of an ink after its deposition on paper. The PE-TMS area/IS area values decreased very rapidly in the first hours with a half-life (fast) value of 0.1 days (±2 h). The solvent evaporation was initially rapid, probably because the resins had not formed a solid film. In the second phase, the evaporation rate decreases considerably with a half-life (slow) value of 9.2 days and can be explained by the resin film formed, which slows down the evaporation process. After that, the PE-TMS area/IS area values were stable at *ca* 60 days, probably due to the change of the resin to a solid film. This phenomenon locked in the ink entries, preventing its evaporation, which allowed the detection of PE-TMS up to 3 years. This exponential two-phase decay ageing behaviour is similar to that already described in the literature [15,16,23]. Koening, Magnolon, and Weyermann (2015) [15] showed that this type of behaviour was observed using four different approaches: (1) when PE was quantified in an ink line, (2) using relative peak area normalising the PE results with stable volatile compounds present in the ink formulation, (3) when determining the solvent loss ratio calculated from PE results obtained by the analyses of naturally and artificially aged samples, and (4) when using a modified solvent loss ratio version calculated from relative peak area results.

In the present study, two programmed independent samples (T(days) = 32 and T(days) = 95) of a black ballpoint pen ink, unknown to the operator (blind data), were also analysed in duplicate on different days to test the reproducibility and accuracy. The results (Figure 2) show that it was possible to correctly estimate the age of deposition of each sample. Although the replicates for 32 days are not very reproducible, both were within the limit of 60 days (defined as the cut-off) showing that the ink was less than 60 days. Regarding the sample for 95 days, it is possible to verify that both replicates present similar results.

The results for PE-TMS blind samples allow for discrimination between inks deposited in the time frame (32 days samples) and out of the frame (95 days samples). However, to improve this timeframe, relative peak areas (RPA) of peaks 1–3 were explored. This approach has already been successfully used by previous authors [5,15,24] and has shown to improve the ink ageing framework. Aginsky (1994) [24] developed a method which allows dating of pen inks without it being necessary to have known ink entries of the same formula than that of the questioned ink. Additionally, with this GC-FID method, the authors proposed that the PE solvent remained at about 20 +/− 10% after 2 years until the age of 15 year due to the polymerisation of resins and other processes, which promotes the decrease of diffusion process, causing the detection. Related to the hardened resins, it is important to note that Aginsky (1994) [24] suggests that in some situations, a “weak” solvent, such as carbon tetrachloride, only extracts the ink volatile compounds from exterior layers, being unable to penetrate inside an old ink. Similar to Lociciro et al., (2004) [5] the present work proposes a mass independent method, in which the RPA are a result of PE-TMS or 1-phenoxypropan-2-ol-TMS peak areas divided by 2-phenoxypropanol-TMS peak area.

The present work data allowed us to successfully establish three dating time frames for the same ink stroke using RPA 3/1 and RPA 2/1 vs. time (days) Figure 3.

The RPA ink ageing curves were also analysed with an exponential two-phase decay. The data show a fast and a slow evaporation region, which corroborates previous data for the PE-TMS aging curve. The first exponential term describes the fast-drying mode, and the second exponential term describes the slow-drying mode. The RPA 3/1 values decreased very rapidly in the first hours with a half-life (fast) value of 0.1 (±3.42 h). The drying rate slowed considerably with a half-life (slow) value of 29 days. After that, the values were stable at 109 days.

Two programmed independent samples (T(days) = 32 and T(days) = 95) of a black ballpoint pen ink, unknown to the operator (blind data), were also fitted to The RPA 3/1 and 2/1 ink ageing curves to test the reproducibility and accuracy. The results show that it was possible to correctly estimate the age of deposition of each sample, where good reproducibility and accuracy at 32 and 95 days were achieved.

Our results show that we successfully developed an ink dating methodology up to 109 days after deposition, using two decay aging curves: PE-TMS and RPA 3/1.

It should be noticed that the present work is a proof of concept where the methodology pretends to contribute to the validation of a non-destructive analytical method of forensic samples.

## 3. Materials and Methods

### 3.1. Chemicals, Materials, and Samples

*N*-hexane (HPLC grade), dichloromethane, and methanol were purchased from Fisher Scientific. Ethyl benzoate and 2-phenoxyethanol (PE) were purchased from Acros Organics. *N*-trimethylsilyl-*N*-methyl trifluoroacetamide (MSTFA) was purchased from Macherey Nagel, and trimethylsilyl chloride (TMCS) was purchased from Alfa Aesar. All solvents used in this study were of high analytical grade (≥99% Reagent).

A black BIC^®^ Crystal Ballpoint Pen was purchased in a commercial area. Ink pen entries were drawn on a white paper from Fegol, multipurpose paper, A4 format (80 g/m^2^). A 1.20 mm diameter (Ø) Harris Uni-Core Punch, purchased from VWR International, was used to cut small paper disc samples. The paper disc samples were placed into sealed tapered vials for analysis.

An ultrasound bath, Branson 2200, and the Thermomixer Compact Eppendorf were used in the extraction procedure. For the derivatisation step, a Bioblock scientific (model number 92721) was used.

### 3.2. Sample Storage

In order to track the behaviour of PE over time, ink samples of Black BIC^®^ ballpoint pen ink were deposited, on white paper, over time on a weekly basis, since 2016. In the case of more recent dates, the deposition was carried out with an interval of 6 h between each record. For each deposition date, ten horizontal strokes were made, about 18 cm long and 1 cm apart. Each sample deposition sheet was separated from the next by two blank sheets. The paper sheets were kept in a closed dossier. The room temperature and humidity were 18–24 °C and 40–60% of relative humidity, respectively, which were close to a standard document storage environment, so no further control was carried out. The conditions of these samples are intended to resemble the real conditions of a questioned ink sample, where the expert is unaware of the document storage conditions to which it was subjected.

### 3.3. Extraction Procedure

For the ink analysis, six samples (of 20 punches) of black ballpoint pen ink were collected for each ink stroke over time. This procedure was done in triplicate for each ink entry and performed by two researchers on different days to obtain independent values. The paper disc samples were placed into sealed conical vials. To prepare blank samples, 20 ink-free paper punches were collected and subjected to the same extraction procedure. The micropunch device was cleaned with n-hexane between each sample extraction to avoid contamination.

The 20 pieces of paper (1.20 mm diameter (Ø) each) were extracted with 63 µL of n-hexane and with 7 µL of internal standard solution (ethyl benzoate in hexane, 500 µg mL^−1^). After sonication for 20 min, the vials were subjected to 1400 rpm at 40 °C (Eppendorf—Thermomixer Compact). Before and after each extraction step, the vials were reserved on ice to decrease the solvents’ volatilisation. After extraction, 25 µL ink solvent samples were derivatised with 25 µL MSTFA:TMCS (95:5, *v*:*v*) at 80 °C for 30 min The absence of non-derivatised compounds peaks was used as the criteria for derivatization reaction completeness. Appendix A depicts the extraction procedure employed in the present study.

### 3.4. GC/MS Conditions

The samples were analysed using an Agilent 6890 Series gas chromatograph coupled with an Agilent 5973 N mass selective detector (Agilent Technologies). A GC MEGA-5 MS (0.25 µm, 0.32 mm, 30 m) capillary column was used for chromatographic separation. The injection volume was 2 µL at 280 °C with a split flow of 5.7 mL/min and a split ratio of 2:1. Helium was used as the carrier gas (1 mL/min). The oven temperature program started at 90 °C (held for 8 min), then ramped up (10 °C/ min) to 100 °C (held for 18 min), and increased to 240 °C at a rate of 30 °C/min (held for 4.67 min) with a transfer line temperature of 280 °C. The ionisation energy was set at 70 eV, and the solvent delay was set at 7 min. The transfer line, MS Source, and MS Quad temperatures were 280, 230, and 150 °C, respectively. The MS analysis was carried out in SCAN (*m/z* 40–250) or SIM mode. Is the latter mode, the monitored ions (*m/z*) for PE were 77, 94, and 138, and PE-TMS were 151, 195, and 210, using 151 as the quantification ion. Ethyl benzoate (internal standard) presented a retention time of 12.5 min, and the chosen quantifier ion was 105 *m/z* in SCAN mode.

### 3.5. Method Calibration

Calibration was performed by analysing standard solutions containing PE-TMS dissolved in n-hexane at different concentrations in quadruplicate: 0.5 µg mL^−1^, 1.0 µg mL^−1^, 2.5 µg mL^−1^, 5.0 µg mL^−1^, 10.0 µg mL^−1^, 20.0 µg mL^−1^, and 50.0 µg mL^−1^. These solutions were prepared by PE-TMS stock solution at 250.0 µg mL^−1^ and by internal standard stock solution.

The ratio peak areas values obtained were analysed and interpreted with the GraphPad Prism 8 (GraphPad Software). The outliers were removed using the Grubbs method. The relative peak area values were obtained by normalising the PE-TMS with the IS peak areas.

### 3.6. Ageing Curve Procedure

For the reproducibility and accuracy assays, two programmed samples (T(days) = 32 and T(days) = 95) of a black ballpoint pen ink, unknown to the operator (blind test), were analysed, and two independent replicates were made to test the reproducibility and accuracy. The ageing curve was generated for PE-TMS by displaying the PE-TMS normalised values as a function of sample age (days). For each ink entry, 6 samples (of 20 punches) were collected. This procedure was done in triplicate, for each known ink age sample entry (T(days) = 0; 0.20; 1; 1.66; 2.66; 3; 5; 14; 33; 55; 109; 182; 365; 1095) and performed by 2 researchers at different days to obtained independent values.

To model the ageing dynamics of the PE-TMS decay, a two-phase exponential function was used due to the fast and slow evaporation region present in the ageing curve, according to the following equation:(1)$$Y=Plateau+SpanFast\;e^{-KFastx}+SpanSlow\;e^{-KSlowx}$$

In which *Plateau* is the *Y* value at infinite times, expressed in the same units as *Y*; *SpanFast* and *SpanSlow* are the starting *Y* values as *Y*0*Fast* and *Y*0*Slow*, and *KFast* and *KSlow* are the two rate constants, expressed in reciprocal of the X axis time units.

The value of *SpanFast* was obtained according to the following equation:(2)SpanFast=Y0×PercentFast×0.01

In which *SpanFast* is the starting *Y* value as *Y*0*Fast*; *Y*0 is the *Y* value when *X* (time) is zero. It is expressed in the same units as *Y*, and *PercentFast* is the fraction of the span (from *Y*0 to *Plateau*) accounted for by the faster of the two components.

The value of *SpanSlow* was obtained according to the following equation:(3)$$SpanSlow=Y0\times \left(100-PercentFast\right)\times 0.01$$

In which *SpanSlow* is the starting *Y* value as *Y*0 *slow*; *Y*0 is the *Y* value when *X* (time) is zero. It is expressed in the same units as *Y*, and *PercentFast* is the fraction of the span (from *Y*0 to *Plateau*) accounted for by the faster of the two components.

A R squared for a confidence interval of 95% was obtained.

## 4. Conclusions

One of the major challenges in forensic document analysis is estimating the age of an ink present on a written document. This work aimed to develop a methodology based on the time solvent evaporation profile to solve real case studies.

The present methodology has been successfully developed and applied to the dating of ballpoint pen inks, where the low levels of PE in ink formulations make it a challenging analyte to detect. The approach novelty is to convert PE to its trimethylsilyl derivative, turning the compound more volatile and amenable to detection in GC-MS analyses, thus increasing the reliability and accuracy of the proposed ink dating method.

A forensic novelty of the proposed approach discriminates three dating time frames: (i) between time 0 and ca 32 days (ageing curve of RPA 2/1), (ii) between time ca 34 and 109 days, and (iii) more than ca 109 days (ageing curve of RPA 3/1). This methodology expands the previously published dating frames for ink ageing profiles [12]. However, it must be stressed that for future forensic applications the number of micro punches must be considerably reduced to preserve the original document.

Furthermore, this methodology should be tested in a larger sample of different black and blue commercial brand pens to establish the validity of the approach to other ink brands. Moreover, validation assays are needed to improve method suitability for trace-level analysis. Finally, research and development in this area can help to improve the reliability and accuracy of ink dating methods, ultimately contributing to the pursuit of justice and the protection of society.

## Figures and Tables

**Figure 1 molecules-28-04781-f001:**
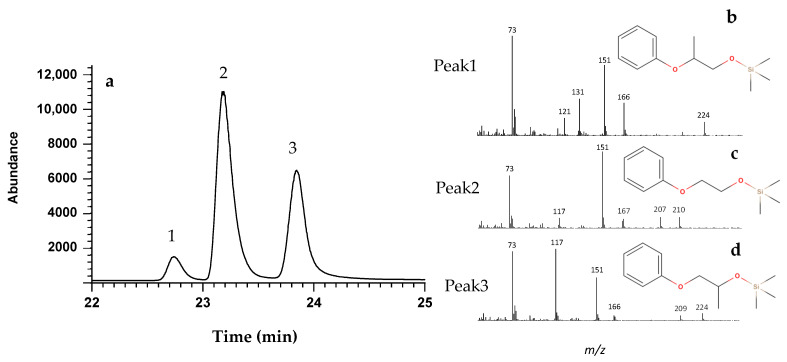
Total ion chromatogram for the black BIC® ballpoint pen ink analysis, focussed on the three main peaks (**a**), as well as the obtained full-scan mass spectrum of peak 1 ((**b**), 2-phenoxypropanol-TMS), peak 2 ((**c**), PE-TMS), and peak 3 ((**d**), 1-phenoxypropan-2-ol-TMS).

**Figure 2 molecules-28-04781-f002:**
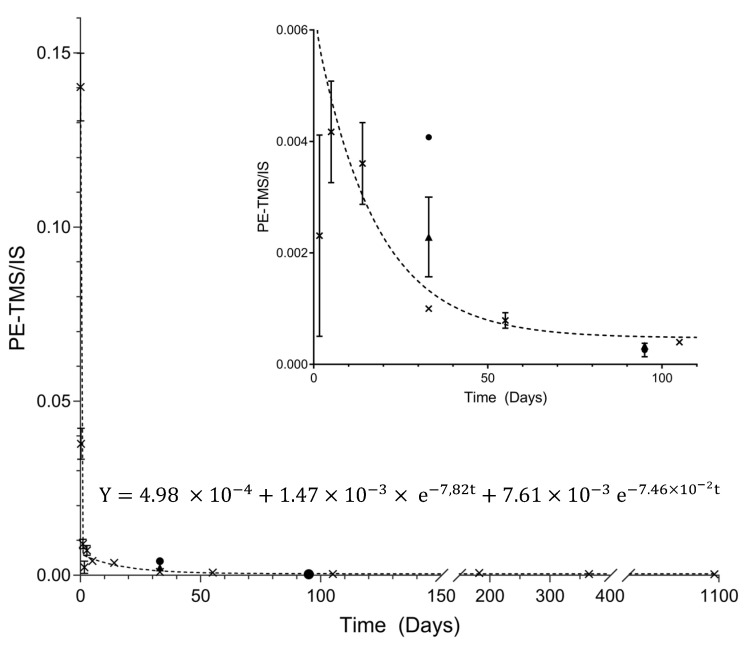
The ageing curve of PE-TMS area/IS area values vs. time (days) obtained from black BIC^®^ ballpoint pen ink fitted by an exponential function two-phase decay. The closed circles and triangles represent the duplicate analysis of two unknown samples to the operator (32 and 95 days). Inset: Expanded fragment of the ageing curve from 0 to 100 days.

**Figure 3 molecules-28-04781-f003:**
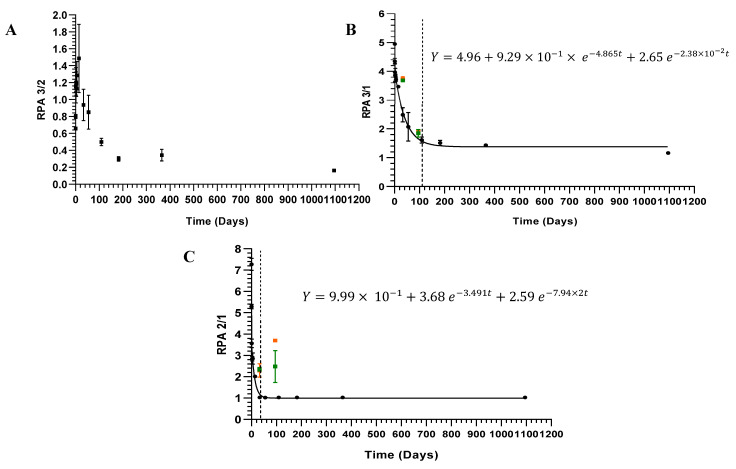
Ageing curves obtained for 0–1095 day old samples from black BIC^®^ ballpoint pen ink: (**A**)—RPA 3/2 vs. time (days); (**B**)—RPA 3/1 vs. time (days); (**C**)—RPA 2/1 vs. time (days). The green and orange dots represent the duplicate analysis of two unknown samples to the operator (32 and 95 days). The dashed line represents the defined as the cut-off of the study: (**B**)—109 days; (**C**)—32 days.

## Data Availability

The data presented in this study are available on request from the corresponding author.

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
