# Peer review of "GC-MS Silylation Derivative Method to Characterise Black BIC® Ballpoint 2-Phenoxyethanol Ratio Evaporation Profile—A Contribution to Ink Ageing Estimation"

_molecules, 2023, doi:10.3390/molecules28124781_

Round 1
Reviewer 1 Report
The submitted manuscript entitled "Solvent ratio evaporation profile of silylation derivatives 2 method to determine black BIC® ballpoint ink ageing" focuses on an important and modern topic as forensic document examination to determine document age and falsification. It should be noted that although a lot of work has been published in this area, up to now there is no ready-made solution, which can be used by forensic institutions without any restriction. Moreover, the methods approved by the courts are usually based on the method of gas chromatography for analysis of volatile components, which allows examination only of documents not older than 2-3 years (which significantly reduces the number of possible documents coming to forensic institutions for expertise). For this reason, the development of new methods and the expansion of existing ones is an important task.The authors proposed a method based on the detection of 2-phenoxyethanol (PE) by GC-MS with further processing and extrapolation of peaks. 2-phenoxyethanol (PE) was chosen as the most common solvent in blue and black ink pens. Despite an interesting approach, many aspects are still unresolved and it is recommended to revise the manuscript significantly in order to make a decision.
- The title should be more precisely defined and specific, otherwise such a title does not make it clear what the manuscript is about;
- Although the manuscript focuses on the use of gas chromatography, in the introduction it is useful to give an overview of existing methods, with a brief description of the advantages and disadvantages. It is important for readers to understand why this methodology was chosen (as there are reasons to doubt the applicability of the methodology with detection of volatile components of ink);
- The authors are invited to extend the introduction (see paragraph above) and to consider additional studies in recent years where alternative approaches based on spectral or other methods have been applied (10.1016/j.dyepig.2016.04.009 ; 10.1021/acs.jcim.9b01037 ; 10.1021/acs.analchem.0c05334 ; 10.1016/j.vibspec.2017.05.006).
- Finally, clear distinctions and comparisons with works in GC-MS should be shown: V. Aginsky and P. Margot. It is also worth pointing out the main differences with the works (This exponential two phase decay aging behaviour is similar to that already described in the literature [16-18]).
- The text and references used should be checked carefully. For example, the text and reference [9] have no connection;
- Some parameters of the experiment (in particular the GC-MS procedure) should be placed in a separate section of SI, as these are technical details;
- At the same time, the methodology of the experiment after the GC-MS is still not sufficiently explained. The basic idea of using the GC-MS results and the post-processing should be shown and explained in detail. What approximations are used to construct the ageing curve, how the validity of this approach can be checked, etc. (см. 10.1016/j.forsciint.2015.03.027);
- In comparable experiments, results for a few inks (same or different colours) are given in order to evaluate the applicability of the technique. The authors of the present manuscript use only one brand of ink. In such a case, it is necessary to explain why this is appropriate;
- In figure 1, the right and left-hand graphs should be marked and also indicated in the caption of the figure.
- The aging curve in Figure 2 looks confusing. Perhaps an expanded fragment of the curve from 0 to 100 days should have been shown additionally. Also, why is the ageing curve shown up to 1000 days? This is really a point worth clarifying.
- The same phrases or repetitions said in other words are often repeated in the discussion and the authors should take this into account. For example,
This approach has already been successfully used by previous authors [17] and has shown to improve the ink ageing framework.
- - The question about the age of the document under study could not be explained. Have such research been carried out? What are the limitations of the proposed method?
Author Response
REVIEWER #1
Letter to Reviewer 1
We want to acknowledge the careful review that contributed to improving the manuscript. The suggestions and comments were very appreciated.
Please, see below our answers to reviewer #1 comments.
- The title should be more precisely defined and specific, otherwise such a title does not make it clear what the manuscript is about;
Reply:
In order to clarify the title, the authors changed to: GC-MS sialylation derivatives method to characterize black BIC® ballpoint 2-phenoxyethanol ratio evaporation profile - a contribution to ink ageing estimation
- Although the manuscript focuses on the use of gas chromatography, in the introduction it is useful to give an overview of existing methods, with a brief description of the advantages and disadvantages. It is important for readers to understand why this methodology was chosen (as there are reasons to doubt the applicability of the methodology with detection of volatile components of ink); The authors are invited to extend the introduction (see paragraph above) and to consider additional studies in recent years where alternative approaches based on spectral or other methods have been applied (10.1016/j.dyepig.2016.04.009 ; 10.1021/acs.jcim.9b01037 ; 10.1021/acs.analchem.0c05334 ; 10.1016/j.vibspec.2017.05.006).
Reply:
Several methods have been described and used to explore ink ageing, and according to a recent INTERPOL report, phenoxythanol is one of the most widely explored compounds for absolute dating. The present method is innovative as it describes a procedure that improves the sensitivity in the analysis of ink ageing curves using gas chromatography-mass spectrometry (GC/MS), through the addition of the derivatization step that allows the monitoring of PE-trimethylsilyl (PE-TMS). Notwithstanding, the authors agree with the suggestions above and enriched the manuscript's introduction with additional methods. The authors also added recent bibliographic references and the references suggested by reviewer #1.
- Finally, clear distinctions and comparisons with works in GC-MS should be shown: V. Aginsky and P. Margot. It is also worth pointing out the main differences with the works (This exponential two phase decay aging behaviour is similar to that already described in the literature [16-18]).
Reply:
As suggested by the reviewer #1 the authors have added the comparisons of the present study with the previous V. Aginsky and P. Margot works. Comparisons were included in the results section.
- The text and references used should be checked carefully. For example, the text and reference [9] have no connection;
Reply:
The text and the references used were carefully checked.
- Some parameters of the experiment (in particular the GC-MS procedure) should be placed in a separate section of SI, as these are technical details;
Reply:
The authors have opted to keep the parameters in the experimental section. The small amount of text does not justify a SI section.
- At the same time, the methodology of the experiment after the GC-MS is still not sufficiently explained. The basic idea of using the GC-MS results and the post-processing should be shown and explained in detail. What approximations are used to construct the ageing curve, how the validity of this approach can be checked, etc. (см. 10.1016/j.forsciint.2015.03.027);
Reply:
The post-processing analysis, namelly the ratio of peaks is a process to minimize manipulation errors, since its mass independent. I tis mentioned at the results with references "Likewise, Lociciro (2004) [5] also refers that the proposed method was mass inde-pendent in which the relative peak area (RPA) was obtained through the area of peak of volatile solvent divided by the area of stable peak (obtained from a dye, a non-volatile compound)."
[5] Lociciro, S.; Dujourdy, L.; Mazzella, W.; Margot, P.; Lock, E. Dynamic of the ageing of ballpoint pen inks: Quantification of phenoxyethanol by GC-MS. Sci. Justice - J. Forensic Sci. Soc. 2004, 44, 165–171, doi:10.1016/S1355-0306(04)71709-8.
- In comparable experiments, results for a few inks (same or different colours) are given in order to evaluate the applicability of the technique. The authors of the present manuscript use only one brand of ink. In such a case, it is necessary to explain why this is appropriate.
Reply:
As stated in a recent INTERPOL report, ink dating remains a challenge where the contribution of different analytical approaches is crucial to achieving a validated method. The present work contributes with a proof of concept to date documents based on PE time profile behaviour, where the SLE extract is derivatized prior to GC/MS analysis. This step can improve method sensitivity to extend the period of PE detectability, distinguish minor differences in PE decay curves and enable the analysis of other compounds that present low volatility or are thermolabile.
- In figure 1, the right and left-hand graphs should be marked and also indicated in the caption of the figure.
Reply:
The figure and figure legend was changed according to reviewer #1 suggestion.
- The ageing curve in Figure 2 looks confusing. Perhaps an expanded fragment of the curve from 0 to 100 days should have been shown additionally. Also, why is the ageing curve shown up to 1000 days? This is really a point worth clarifying.
Reply:
We agree with the reviewer, and an expanded fragment was added. The analysis was carried out for up to 1000 days to verify the decay's stabilization.
- The same phrases or repetitions said in other words are often repeated in the discussion and the authors should take this into account. For example, This approach has already been successfully used by previous authors [17] and has shown to improve the ink ageing framework.
Reply:
The text was revised according to reviewer #1 suggestions and repeated ideas were removed.
- The question about the age of the document under study could not be explained. Have such research been carried out? What are the limitations of the proposed method?
Reply:
As suggested by reviewer #1, the method limitations were added to the conclusion section
Reviewer 2 Report
The article is based on the optimization of a method for determining the age of inks using phenoxyethanol. Although real samples from one of the most widely used pens in the market (Bic) were used, phenoxyethanol has been extensively studied by scientific research, and countless studies have shown that this compound cannot date inks beyond 4 months because it does not continue to change over time.
Furthermore, in the forensic field, it is not possible to use a method that requires 20 microperforations for the analysis of ink data, as current research is working with 4 microperforations using a much simpler extraction procedure. Working with 20 microperforations means almost completely destroying the forensic sample, and it is not permitted by current legislation. These two points are important to be reflected in the article if it is to be published.
However, the method used is novel and not described in the abundant literature. Specifically, solid-liquid extraction with n-hexane in the presence of an internal standard (ethyl benzoate) followed by derivatization with a silylation reagent, analyzing the aging curve using gas chromatography-mass spectrometry (GC/MS) to monitor the PE-trimethylsilyl (PE-TMS), in order to improve the method's sensitivity.
Another novelty is that, due to the fact that this method does not allow the detection of phenoxyethanol after three years because silylation increases volatility, and in currently used methodologies, phenoxyethanol is still detected in inks many years after their deposition on paper. This can be interesting in order to establish a cut-off point after three years to assert that Bic pen inks are not more than three years old, as well as to allow for relative dating in three different time frames.
MINOR REVISIONS: • Add a table reflecting the initial concentration of the monitored compound in black Bic pen ink, as it is suggested to be an ideal method when PE concentrations are low.
• The bibliography should be updated to more recent publications, and it is recommended to review the latest Interpol report: https://www.interpol.int/content/download/14458/file/Interpol%20Review%20Papers%202019.pdf
• In the conclusions, it should be emphasized that the methodology still needs improvement by working with fewer microperforations since 20 microperforations are not compatible for working with forensic samples. Furthermore, this type of methodology should be experimentally tested with blue Bic pens and pens from other brands, especially since black pens are the least used in forensic samples.
• The conclusions should be limited to a dating model for the analyzed pen ink and not extended to all pen inks, as the different chinetikinetics in pen inks of different chemical compositions have been observed.
Author Response
REVIEWER #2
Letter to Reviewer 2
We want to acknowledge the careful revision of reviewer #2, which contributes to improving the manuscript.
Please, see below our answers to reviewer #2 comments.
- The article is based on the optimization of a method for determining the age of inks using phenoxyethanol. Although real samples from one of the most widely used pens in the market (Bic) were used, phenoxyethanol has been extensively studied by scientific research, and countless studies have shown that this compound cannot date inks beyond four months because it does not continue to change over time. Furthermore, in the forensic field, it is not possible to use a method that requires 20 microperforations for the analysis of ink data, as current research is working with 4 microperforations using a much simpler extraction procedure. Working with 20 microperforations means almost completely destroying the forensic sample, and it is not permitted by current legislation. These two points are important to be reflected in the article if it is to be published.
Reply:
As suggested by reviewer #2 both points were addressed in the manuscript discussion and in the conclusions.
- However, the method used is novel and not described in the abundant literature. Specifically, solid-liquid extraction with n-hexane in the presence of an internal standard (ethyl benzoate) followed by derivatization with a sialylation reagent, analyzing the aging curve using gas chromatography-mass spectrometry (GC/MS) to monitor the PE-trimethylsilyl (PE-TMS), in order to improve the method's sensitivity. Another novelty is that, due to the fact that this method does not allow the detection of phenoxyethanol after three years because sialylation increases volatility, and in currently used methodologies, phenoxyethanol is still detected in inks many years after their deposition on paper. This can be interesting in order to establish a cut-off point after three years to assert that Bic pen inks are not more than three years old, as well as to allow for relative dating in three different time frames.
Reply:
We deep appreciate reviewers #2 comments.
MINOR REVISIONS:
- Add a table reflecting the initial concentration of the monitored compound in black Bic pen ink, as it is suggested to be an ideal method when PE concentrations are low.
Reply:
The method developed does not depends directly on the compound mass, since rely on a ration between compounds. Thus the accurate determination of concentration was not taken in account. Consequently, a table withe concentrations can not be drawn.
- The bibliography should be updated to more recent publications, and it is recommended to review the latest Interpol report: https://www.interpol.int/content/download/14458/file/Interpol%20Review%20Papers%202019.pdf
Reply:
Accordingly, to the reviewer's suggestion, the references were updated, and the manuscript's introduction was expanded to include work from recent publications.
- In the conclusions, it should be emphasized that the methodology still needs improvement by working with fewer microperforations since 20 microperforations are not compatible for working with forensic samples. Furthermore, this type of methodology should be experimentally tested with blue Bic pens and pens from other brands, especially since black pens are the least used in forensic samples.
Reply:
We agree with the reviewer's suggestion. The conclusion was expanded to include the suggestions.
- The conclusions should be limited to a dating model for the analyzed pen ink and not extended to all pen inks, as the different chinetikinetics in pen inks of different chemical compositions have been observed.
Reply:
The authors agree with the clarification of methodology improvements and integrated the above limitations in the conclusion section
Round 2
Reviewer 1 Report
The authors have revised the manuscript significantly and responded to most of the questions, however, some points remain to be explained.
- It is recommended to present the experimental steps in the form of illustrations or diagrams. There are quite a lot of such examples in the literature. This will make it much easier for the readers to understand.
Although the authors describe this work as a Concept, it is not superfluous to demonstrate the efficiency of the technique on "test" samples or, at least, to make speculations about the applicability of the proposed approach to the analysis of real objects and documents.
It is not possible to estimate the caption to figure 1, as there was an "overlapping" of text and image in the manuscript version submitted. But it is still recommended to mark (a,b,c...) the right and left figure and to indicate in the legend, as there are at least four Spectra in this image.
The authors have made a slight improvement and added some points for later times (X-axis). However, the detailing of the time period from day 0 to 100 has not yet been carried out, as this is where significant changes are occurring.
Author Response
REVIEWER #1 (SECOND ROUND)
Letter to Reviewer 1
We want to acknowledge the careful review that contributed to improving the manuscript. The suggestions and comments were very appreciated.
Please, see below our answers to reviewer #1 (round 2) comments.
- It is recommended to present the experimental steps in the form of illustrations or diagrams. There are quite a lot of such examples in the literature. This will make it much easier for the readers to understand.
Reply:
As suggested by the reviewer #1 the authors have added an illustration of the experimental steps as supplementary material.
- Although the authors describe this work as a Concept, it is not superfluous to demonstrate the efficiency of the technique on "test" samples or, at least, to make speculations about the applicability of the proposed approach to the analysis of real objects and documents.
Reply:
The technical efficiency was tested using two programmed independent samples [T(days)=32 and T(days)=95] of a black ballpoint pen ink unknown to the operator (blind data), where acceptable reproducibility and accuracy were achieved for RPA 3/1.
- It is not possible to estimate the caption to figure 1, as there was an "overlapping" of text and image in the manuscript version submitted. But it is still recommended to mark (a,b,c...) the right and left figure and to indicate in the legend, as there are at least four Spectra in this image.
Reply:
As suggested by the reviewer #1 the authors have corrected Figure 1 and the caption.
- The authors have made a slight improvement and added some points for later times (X-axis). However, the detailing of the time period from day 0 to 100 has not yet been carried out, as this is where significant changes are occurring."
Reply:
As suggested by the reviewer #1 the authors have improved the figure, detailing the time period from day 0 to 100.